# Iron Oxide Nanoparticles with Supramolecular Ureido-Pyrimidinone Coating for Antimicrobial Peptide Delivery

**DOI:** 10.3390/ijms241914649

**Published:** 2023-09-27

**Authors:** Chiara Turrina, Jennifer Cookman, Riccardo Bellan, Jiankang Song, Margret Paar, Patricia Y. W. Dankers, Sonja Berensmeier, Sebastian P. Schwaminger

**Affiliations:** 1Chair of Bioseparation Engineering, School of Engineering and Design, Technical University of Munich, Boltzmannstr. 15, 85748 Garching, Germany; c.turrina@tum.de (C.T.);; 2Department of Chemical Sciences, Bernal Institute, University of Limerick, V94 T9PX Castletroy, Ireland; jennifer.cookman@ul.ie; 3Department of Biomedical Engineering, Institute for Complex Molecular Systems, Eindhoven University of Technology, 5600 MB Eindhoven, The Netherlands; r.bellan@tue.nl (R.B.);; 4Division of Medicinal Chemistry, Otto Loewi Research Center, Medical University of Graz, Neue Stiftingtalstraße 6, 8010 Graz, Austria; 5BioTechMed-Graz, Mozartgasse 12, 8010 Graz, Austria

**Keywords:** iron oxide nanoparticles, ureido-pyrimidinone, supramolecular system, antimicrobial peptide, intracellular delivery, cytocompatibility

## Abstract

Antimicrobial peptides (AMPs) can kill bacteria by disrupting their cytoplasmic membrane, which reduces the tendency of antibacterial resistance compared to conventional antibiotics. Their possible toxicity to human cells, however, limits their applicability. The combination of magnetically controlled drug delivery and supramolecular engineering can help to reduce the dosage of AMPs, control the delivery, and improve their cytocompatibility. Lasioglossin III (LL) is a natural AMP form bee venom that is highly antimicrobial. Here, superparamagnetic iron oxide nanoparticles (IONs) with a supramolecular ureido-pyrimidinone (UPy) coating were investigated as a drug carrier for LL for a controlled delivery to a specific target. Binding to IONs can improve the antimicrobial activity of the peptide. Different transmission electron microscopy (TEM) techniques showed that the particles have a crystalline iron oxide core with a UPy shell and UPy fibers. Cytocompatibility and internalization experiments were carried out with two different cell types, phagocytic and nonphagocytic cells. The drug carrier system showed good cytocompatibility (>70%) with human kidney cells (HK-2) and concentration-dependent toxicity to macrophagic cells (THP-1). The particles were internalized by both cell types, giving them the potential for effective delivery of AMPs into mammalian cells. By self-assembly, the UPy-coated nanoparticles can bind UPy-functionalized LL (UPy-LL) highly efficiently (99%), leading to a drug loading of 0.68 g g^−1^. The binding of UPy-LL on the supramolecular nanoparticle system increased its antimicrobial activity against *E. coli* (MIC 3.53 µM to 1.77 µM) and improved its cytocompatible dosage for HK-2 cells from 5.40 µM to 10.6 µM. The system showed higher cytotoxicity (5.4 µM) to the macrophages. The high drug loading, efficient binding, enhanced antimicrobial behavior, and reduced cytotoxicity makes ION@UPy-NH_2_ an interesting drug carrier for AMPs. The combination with superparamagnetic IONs allows potential magnetically controlled drug delivery and reduced drug amount of the system to address intracellular infections or improve cancer treatment.

## 1. Introduction

The 17 global grand challenges stated in the United Nations Sustainable Development Goals target good health and well-being. One of the goals within this challenge is to address the growing resistance of pathogenic bacteria to conventional antibiotics [1,2,3]. A newly developed therapeutic option is the use of antimicrobial peptides (AMPs) that appear in various life forms, have a broad spectrum of antimicrobial activity, and differ from traditional antibiotics’ mode of action [4,5,6,7]. Most AMPs are cationic and contain between 10 and 60 amino acids [8]. The specific targeting of the positively charged peptides is based on the cell membrane composition. The bacterial cell envelope contains negatively charged components like phosphatidylserine, whereas human cell membranes are based on neutral units such as phosphatidylcholine. The AMPs interact electrostatically with the negatively charged membranes and accumulate there. This behavior induces higher permeability, membrane lysis, release of intracellular components, and, therewith, cell death [7,8]. AMPs are highly specific and have a low tendency to trigger antimicrobial resistance [7]. The best-analyzed group of AMPs comprises linear, amphiphilic, α-helical peptides [9]. One of these is lasioglossin III (LL, H-Val-Asn-Trp-Lys-Lys-Ile-Leu-Gly-Lys-Ile-Ile-Lys-Val-Val-Lys-NH_2_), a part of the venom of the bee Lasioglossum laticeps [10]. It shows high antimicrobial activity against Gram-positive and Gram-negative bacteria, e.g., *B. subtilis*, *S. aureus*, and *E. coli* [10]. Furthermore, it has anticancer activity against leukemia cells and solid tumors like PC12, HeLa S3, and SW480 and has a low hemolytic potential [10]. Though AMPs show multiple advantageous properties, their application in the human body is often limited due to stability and toxicity problems [4,7]. Therefore only a few AMPs are already approved by the FDA or are in the clinical stage [4,11,12]. Effective drug delivery can help overcome high pharmacological doses that result in toxic side effects [13,14]. Both supramolecular engineering and nanoparticles as drug carriers have the potential to improve the behavior of AMPs, for example, by reducing cytotoxicity and enhancing antimicrobial behavior [15,16,17,18,19].

Supramolecular engineering is the deliberate design and assembly of molecular structures and materials through noncovalent interactions to create functional systems with specific properties. Prominent examples are the self-assembly of DNA, host–guest chemistry, and supramolecular polymers [20,21]. A supramolecular AMP assembly is a modular and tunable approach, allowing the flexible incorporation of functionalized molecules to generate multifunctional systems [22,23,24]. The ureido-pyrimidinone (UPy) unit is a common monomer that is connected to a urea group and a hydrophobic alkyl spacer linked with a water-soluble oligo (ethylene glycol) by a urethane unit. The UPy unit can form amphiphilic supramolecular systems based on a fourfold hydrogen bonding approach [25]. It can form hydrogen-bonded dimers protected by hydrophobic pockets [26]. The UPy dimers can laterally stack and assemble into fibrous structures [26,27]. LL can be functionalized with a UPy unit and self-assembled with other UPy moieties like amine-functionalized UPy (Figure 1) [28]. Song et al. formed UPy–AMP assemblies by self-assembly and with controlled antimicrobial activity. Furthermore, they influenced the cytocompatibility by adjusting the AMP concentration [28].

Nanotechnology has the potential to reinvent medicine in the future, leading to improved diagnosis of, therapy for, and drug delivery for diseases [29,30]. Combining supramolecular engineering with iron oxide nanoparticles (IONs) enables a magnetically controlled drug delivery system [31,32]. IONs have the potential to carry a large drug dose due to the high surface area compared to its volume and can be guided by an external magnetic field directly to a target. Therefore, a high local concentration can be reached while lower drug amounts must be incorporated into the human body, which reduces toxicity and side effects [32,33,34,35]. Furthermore, IONs can be visualized by magnetic resonance imaging (MRI) and administered for hyperthermia patients by alternating the magnetic field to generate heat [36,37,38]. A fast and cost-efficient synthesis method used to manufacture the IONs is coprecipitation using the Massart process [17,39]. In this method, superparamagnetic iron oxide nanoparticles ranging in size between 4 and 16 nm and with a high specific surface area >100 m^2^ g^−1^ are generated [40,41,42,43]. The combination of bare IONs with LL showed improved antimicrobial activity yet led to uncontrolled agglomeration and a nonspecific weak binding with potential loss of the AMP [17].

The usage of UPy-coated IONs can combine the advantages of supramolecular engineering like efficient binding by self-assembly, adjustable cytocompatibility, and a controlled magnetic drug delivery with enhanced antimicrobial activity [44]. We showed the synthesis route and characterization (infrared spectroscopy, X-ray diffraction, thermogravimetric analysis, dynamic light scattering, zeta potential, Nile red assay, and superconducting quantum interference device) of ION@UPy-NH_2_ in a previous work [44]. Bare IONs (BIONs) synthesized by the Massart process built the magnetic core of ION@UPy-NH_2_ with a d_TEM_ of 8.74 ± 1.61 nm [15]. After a two-step functionalization with polyglutaraldehyde (PGA), UPy-NH_2_ can interact with the particles via imine binding [44]. ION@UPy-NH_2_ was previously characterized by Turrina et al. as showing a crystalline iron oxide core with superparamagnetic behavior, saturation magnetization around 31 Am^2^ kg^−1^, a hydrodynamic diameter of 177 nm (pH 7), and a positive surface charge [44]. An amount of 7.23 wt% of UPy-NH_2_ was bound, and Nile red assay indicated the lateral UPy stacking [44]. Yet, the particle’s morphology, cytocompatibility, internalization behavior, and interaction with a potential drug have not been analyzed.

This study aims to add to the knowledge of ION@UPy-NH_2_ as a potential supramolecular-based, magnetically controllable drug carrier for LL. Its morphology was analyzed by combining high-angle annular dark field scanning transmission electron microscopy (HAADF-STEM), integrated differential phase contrast (iDPC), and cryo-TEM imaging. Its biological performance, including antimicrobial activity, cytocompatibility, and internalization with THP-1-derived macrophages as phagocytic cells and human kidney cells (HK-2) as nonphagocytic cells of the drug carrier alone and in combination with UPy-LL, was analyzed.

## 2. Results and Discussion

### 2.1. ION@UPy-NH_2_

The multiple-step synthesis route is described in a previous publication of Turrina et al. [39]. The bare IONs were synthesized by coprecipitation. For the UPy coating, first, the particles were functionalized with (3-Aminopropyl)triethoxysilan (APTS), followed by a poly(glutaraldehyde) (PGA) coating, and finally a coating with UPy-NH_2_ [39].

#### 2.1.1. Particle Morphology

The morphology of ION@UPy-NH_2_ was analyzed with a combination of different TEM techniques. Imaging with HAADF-STEM provided more insights into the composition and shape of the ION@UPy-NH_2_ (Figure 2). In combination with DPC imaging, the crystalline nanoparticle core (green circle), showing atomic lattice planes, can be differentiated from the amorphous supramolecular core. Moiré fringes were also observed (Figure 2, red outline), suggesting that a smaller crystalline particle is present, causing a lattice plane overlap, creating the larger planes known as Moiré fringes. Due to the size and low z-contrast of the combined nanoparticle construct, HAADF-STEM does not allow for easy distinction between the ION core and the biomolecular coating. By using an alternative imaging technique known as integrated differential phase contrast imaging, this distinction can be made. The iDPC-STEM imaging technique takes advantage of the phase signal resulting from electron beam interactions with the sample. In magnetic materials such as IONs, the transmitted phase signal is drastically deflected compared to the nonmagnetic coating. The coating showed sensitivity to the scanning electron beam in STEM mode (Figure 2b,c, red arrow).

DPC allowed lower electron doses to be used while acquiring higher-resolution images (Figure 2). In the HAADF-STEM, atomic lattice planes could not be clearly identified, but when the DPCx and DPCy images were observed, atomic lattice planes and even atomic resolution were visible. Figure 3 shows the agglomerate where the iron oxide core can be identified with respect to the amorphous exterior.

Cryo-TEM analysis indicated the formation of ~1 µm (928 nm ± 128 nm) sized, UPy-NH_2_-based fibers that could not be visualized with the former techniques (Figure 4a,b) [45]. Two fibers interact directly with the ION@UPy-NH_2_ agglomerate, while one appears to be loose, which could be induced by the preparation. UPy-NH_2_ binds on the surface and forms a core–shell structure. The fibers can either grow from the surface or form independently and complex afterward with the particles. Because after synthesis, the particles were ultrasonicated and intensely washed (7×) by magnetic decantation, we hypothesize a binding. Previous Nile red assay showed a huge shift from approximately 656 nm to a lower wavelength of 632 nm, confirming UPy-NH_2_ fiber formation [44].

#### 2.1.2. Cytocompatibility

The cytocompatibility of ION@UPy-NH_2_ and its precursors ION@PGA and BIONs were analyzed quantitatively and qualitatively at various concentrations. Inside the body, the particles meet phagocytic cells and nonphagocytic cells; therefore, the effect of the particles’ interactions on HK-2- and THP-1-derived macrophages was investigated by resazurin assay and, additionally, live/dead staining (Figure 5) [46]. The cell viability assay uses the reducing agent nicotinamide adenine dinucleotide (NADH) as an electron source to transform resazurin and induce a color shift [47]. Therefore, it can be used as a marker for metabolic activity [47]. Calcein-AM is used as green staining for the entire living cell. Propidium iodide stains the dead cells red [48]. For HK-2 cells, all investigated particles (concentrations between 0.03 g L^−1^ and 0.50 g L^−1^) showed cell viability greater than 70%, a threshold value for cytocompatibility (ISO-10993, Figure 5a). Good cell viability for BIONs and IONs with various coatings, e.g., folic acid and carboxymethyl dextran (CMD), was previously reported for multiple cell lines [44,49]. In contrast, resazurin experiments with THP-1 cells demonstrated concentration-dependent cytotoxicity for all particles (Figure 5b). Particle concentrations higher than 0.30 g L^−1^ reduced the metabolic activity by around 53%. It is hypothesized that high particle concentrations embody a nonideal environment for the THP-1 cells, where the conditions are unfavorable for proliferation and the metabolic activity is reduced while the cells remain alive. This state of reversible cell cycle arrest is called quiescence [50]. Similar behavior was analyzed by Fernandes et al., who used doxorubicin-loaded nanocubes in combination with magnetic hyperthermia to induce low proliferation quiescence to cancer stem cells [51]. Furthermore, IONs can release free iron to the cytoplasm when endosomes or lysosomes transport them under acidic conditions [52]. Toxic reactive oxygen species (ROS) can be generated by free radical formation due to the Fenton reaction [53]. Kim et al. showed that dextran-coated IONs increased ROS production in hematopoietic stem cells [54,55].

After the drug carrier and its precursors showed good cytocompatibility to both cell types at various concentrations, the next step is to check if internalization is taking place.

#### 2.1.3. Internalization

The ION@UPy-NH_2_ shape is a combination of spherical nanoparticles and UPy fibers as evidenced by the cryo-TEM analysis in Figure 4. For internalization experiments,

Endocytosis is a dynamic and versatile internalization process that leads to the internalization of extracellular IONs [56]. IONs can nonspecifically adsorb on the cell surface by electrostatic interaction, which induces the activation of endocytic mechanisms and leads to internalization [57]. Also, for macrophagic cells (RAW264.7), it was shown that endocytic pathways, including macropinocytosis or phagocytosis, internalize IONs [58]. Specifically, positively charged IONs are attracted by the negatively charged phospholipids and proteins on the plasma membrane [57]. Our experiments did not allow us to identify the internalization mechanism of the here-synthesized materials conclusively.

ION@UPy-NH_2_ were labeled with UPy-Cy5 (Appendix A) to generate red fluorescent particles. HK-2 cells showed complete adsorption of the particles on the cellular envelope within 120 min (Figure 6). After 24 h, the visible particles were internalized. Size shape and surface charge highly impact internalization [59]. In THP-1 macrophages, ION@UPy-NH_2_ adsorbed in the first 5 min and showed a high degree of internalization after 2 h, which increased within 24 h. While spherical particles were internalized within 2 min, wormlike particles were not internalized in the first 30 min and showed a lower degree of internalization after 22 h [60]. Champion et al. analyzed the internalization of differently shaped particles into alveolar macrophages by phagocytosis [60].

The cell experiments demonstrated good cytocompatibility (>70%) of ION@UPy-NH_2_ for HK-2 cells and potential toxicity for macrophagic cells (THP-1) at high concentrations. The particles were internalized in both cells types. Because of these aspects, ION@UPy-NH_2_ represents a promising new drug carrier system.

### 2.2. Binding of UPy-LL to ION@UPy-NH_2_

The supramolecular interaction caused by hydrogen bonding of UPy units allows the addition of UPy-based drug molecules and generates an innovative magnetically controlled drug delivery system.

The binding of UPy-LL on ION@UPy-NH_2_ was analyzed in binding experiments. UPy-LL can self-assemble with UPy-NH_2_ [28]. After magnetic decantation, the supernatant was photometrically analyzed at 280 nm, and the bound UPy-LL amount was calculated. The binding was highly efficient (Figure 7a). To find the ideal binding conditions, different particle and UPy-LL concentrations were analyzed. Independent of the particle concentration, 99% of UPy-LL (<0.5 g L^−1^) was bound. Compared to the binding of bare IONs with LL, where a maximal loading of 0.23 g g^−1^ LL on bare IONs with a starting concentration of 2 g L^−1^ was reached (efficiency: 11.5%), the supramolecular engineering highly improved the binding efficiency (99%) [17]. It is hypothesized that saturation occurs because of sterical hindrance due to the α-helical shape of LL. The highest reached drug loading was 0.68 g UPy-LL per gram ION@UPy-NH_2_ (Appendix A). Nile red assay demonstrated a blue shift from 656 nm to 638 nm by increasing the UPy-LL amount (Figure 7b). The shift indicates the formation of UPy-LL or UPy-NH_2_-UPy-LL fibers. Cryo-TEM imaging of ION@UPy-NH_2_@UPy-LL showed particle agglomerates with directly bound short fibers with a size of 135 ± 55 nm (Figure 7c). The following experiments were conducted with ION@UPy-NH_2_@UPy-LL in which the particle concentration during adsorption was 0.4 g L^−1^.

The high binding efficiency allows the generation of a drug delivery system with high drug loading. Antimicrobial tests can be used to analyze if the drug is still active while being bound to the drug carrier.

#### 2.2.1. Antimicrobial Activity

The antimicrobial properties of ION@UPy-NH_2_, ION@UPy-NH_2_@UPy-LL, and free UPy-LL were analyzed by culturing these materials with a green fluorescent protein (GFP)-expressing *E. coli* (BL21, ampicillin resistance). Therefore, the fluorescent bacteria were counted after 24 h. ION@UPy-NH_2_ without AMP led to comparable growth of the negative control (cells that were not combined with particles or AMP) for concentrations ≤ 0.03 g L^−1^ (Figure 7c). Higher particle concentrations induced less growth with 28% viability. Previous experiments with BIONs showed a reduction in *E. coli* growth at high particle concentrations of around 68% viability [17]. The stronger decrease could come from the positive ION@UPy-NH_2_ surface being more attracted to the negative bacteria surface. Free LL-III has a minimum inhibitory concentration of 1.13–3.7 µM for *E. coli* [10,17,19]. In our experiments, the UPy-LL led to nearly no bacteria growth at 1.77 µM and complete inhibition at 3.53 µM (Figure 7d). Bound to ION@UPy-NH_2,_ the antimicrobial behavior is slightly improved, leading to a MIC of 1.77 µM (0.02 g L^−1^ particle concentration). Measurements of the growth with OD600 confirm these data (Appendix A). Because of the low particle concentration, the effect of ION@UPy-NH_2_ can be neglected. A similar effect of bound LL showing higher antimicrobial activity was shown by Turrina et al. for BION@LL and ION@CMD@LL (adsorbed) [15,17]. Because of the efficient UPy-LL binding, low particle concentrations can generate the highly efficient antimicrobial behavior of ION@UPy-NH_2_@UPy-LL.

#### 2.2.2. Cytocompatibility

The cationic nature of AMPs is likely to make them highly antibacterial but also bears potential cytotoxicity against mammalian cells, limiting their successful usage [4,61,62]. Similar to ION@UPy-NH_2,_ the cytocompatibility of ION@UPy-NH_2_@UPy-LL and free UPy-LL was investigated (Figure 8). UPy-LL showed cell viability >70% until 5.4 µM for HK-2 and 3.53 µM for THP-1 in resazurin assay (Figure 8a,b). Live/dead staining showed a similar trend: UPy-LL was more cytotoxic for the macrophagic THP-1 cells than for HK-2 (Figure 8c,d). For both resazurin assay and live/dead staining, nearly no living cells were left at 14.1 µM for HK-2 and 10.6 µM for THP-1. The resazurin data show ION@UPy-NH_2_@UPy-LL led to good cytocompatibility (>70%) until 10.6 µM (0.12 g L^−1^ particle concentration) for HK-2 cells and 3.53 µM for THP-1-derived macrophages (0.04 g L^−1^ particle concentration). At the respective particle concentration, no cytotoxic effect was measured for ION@UPy-NH_2_ (Section 2.1.2). Especially for higher concentrations of UPy-LL, the combination with the particles improves the cytocompatible behavior for both HK-2 and THP-1. At 14.1 µM of bound UPy-LL, metabolic activity is 50.4% HK-2 and 34.7% THP-1.

The cytotoxic effect of UPy-LL is impeded by binding to the particles, and ION@UPy-NH_2_@UPy-LL is less cytotoxic towards HK-2 cells than towards THP-1 cells (Figure 8a,b). Figure 8c,d illustrate differences in live and dead staining for different lasioglossin concentrations. Čeřovský et al. observed toxicity against rat epithelial cells (IEC-6) of 19 µM for free LL [10].

## 3. Materials and Methods

We showed the detailed three-step synthesis route and characterization (infrared spectroscopy, X-ray diffraction, thermogravimetric analysis, dynamic light scattering, zeta potential, Nile red assay, and superconducting quantum interference device) of ION@UPy-NH_2_ in a previous work [39]. The bare IONs were synthesized by coprecipitation. For the UPy coating, first, the particles were functionalized with (3-Aminopropyl)triethoxysilan (APTS), followed by a poly(glutaraldehyde) (PGA) coating, and finally a coating with UPy-NH_2_ [39]. ION@UPy-NH_2_ showed superparamagnetic behavior with a saturation magnetization of 31 Am^2^ kg^−1^, a positively charged surface, and a hydrodynamic diameter of 177 nm at pH 7. Thermogravimetric analysis determined that 7.23 wt% of the overall particle weight is the bound Upy-NH_2_. Nile red assay showed a shift to lower wavelength (631 nm at pH 7), indicating the formation of hydrophobic pockets [44].

### 3.1. Morphology

High-angle annular dark field scanning transmission electron microscopy (HAADF-STEM) and integrated differential contrast (iDPC) STEM imaging were conducted on a Thermo Fisher Scientific Titan Themis Cubed operating at an acceleration voltage of 300 kV and tuned with a monochromator and probe corrector. HAADF-STEM and iDPC imaging were acquired using Velox (Thermo Fisher Scientific, Dreieich, Germany) software. For sample preparation, a Lacey carbon 200 mesh copper grid (Agar Scientific, Stansted, UK) was plasma-treated using a Gatan Solarus 950 Advanced plasma system (Gatan Inc., Berwyn, PA, USA) with O_2_ for 30 s at 65 W to induce a hydrophilic surface on the grid. The TEM grid was suspended using reverse-action tweezers, and using a micropipette, a 7 µL aliquot of 100× diluted sample was deposited on the grid and left under cover until the droplet was evaporated. Furthermore, the grid was kept under a high vacuum overnight, ensuring complete evaporation and minimizing imaging artifacts. For the DPC measurement the detector is segmented into different areas (A–D). The sample can deflect the electronic beam. Such deflection can create images where the A-C (DPCx (A–C)) and B-D (DPCy (B–D)) detector segments are differentiated. Cryo-TEM imaging was performed on samples with 0.5 g L^−1^ particles and incubated overnight in Millipore water using quantifoil carbon-covered grids (Electron Microscopy Sciences, Hatfield, PA, US 200 mesh, 50 µm hole size). Before sample addition, grids were surface-plasma-treated (at 5 mA for 40 s) using a Cressington 208 carbon coater. Using an automated vitrification robot (FEI Vitrobot™ Mark III, Hillsboro, OR, US) operating at 22 °C and a relative humidity of 100%, a 3 µL sample was applied to the grids. The excess sample was removed using blotting filter paper for 3 s at −3 mm. The thin film formed was vitrified by plunging the grid into liquid ethane and subsequently, liquid nitrogen. The vitrified grid was transferred to a cryo-transfer holder and prepared for TEM imaging. TEM imaging was conducted using a CryoTITAN equipped with a field emission gun operating at an acceleration voltage of 300 kV, a postcolumn Gatan energy filter, and a 2048 × 2048 Gatan CCD camera. Vitrified films were observed with the CryoTITAN microscope at temperatures below −170 °C. Micrographs were taken at low-dose conditions using a defocus setting of −5 µm or −1 µm at 25 k magnification.

### 3.2. Cell Culture

For cell culture, Dulbecco’s Modified Eagle Medium (DMEM (1×), gibco, ref. 42430-025) and Roswell Park Memorial Institute 1640 (RPMI (1×), gibco, ref. A10491-01) were supplemented with 10% fetal bovine serum (FBS) and 1% penicillin/streptomycin (P/S), respectively. Human kidney cells (HK-2, passaged 2× per week) and monocytic human THP-1 cells (ATCC, passaged every 2nd day) were cultured at 37 °C in 95% air/5% CO_2_ atmosphere with DMEM and RPMI medium, respectively. The differentiation of THP-1 monocytes into macrophages was induced by adding 50 ng/mL phorbol 12-myristate 13-acetate (PMA) to the culture medium and incubated for 48 h. The experiments were carried out with a seeding density of 2.5 × 10^5^ cells/cm^2^ for the THP-1 cells and 2.5 × 10^4^ cells/cm^2^ for the HK-2 cells.

### 3.3. Cytocompatibility

The cytocompatibility of BIONs, ION@PGA, ION@UPy-NH_2_, ION@UPy-NH_2_@UPy-LL (preparation can be found in 2.6 UPy-LL binding), and free UPy-LL for THP-1 macrophages and HK-2 cells was investigated by resazurin assay. The cells were seeded onto a 96-well plate (*n* = 3). The THP-1 cells were induced with PMA for 48 h, and the HK-2 cells were cultured overnight to allow cell adhesion. The particles were suspended in 70% EtOH by magnetic decantation and sterilized for two hours under UV light. Afterward, the supernatant was changed to PBS by magnetic decantation, removing the EtOH supernatant and resuspending the nanoparticles in PBS. The IONs were then ultrasonicated for 30 min in an ultrasonic bath to ensure full resuspension. The respective medium exchanged the buffer. After attachment, the culture medium was exchanged for a medium containing BIONs, ION@PGA, and ION@UPy-NH_2_ at the concentrations of 0 g L^−1^, 0.03 g L^−1^, 0.05 g L^−1^, 0.08 g L^−1^, 0.10 g L^−1^, 0.30 g L^−1^, and 0.50 g L^−1^, respectively, or ION@UPy-NH_2_@UPy-LL and UPy-LL at 0 µM, 1.77 µM, 3.53 µM, 5.40 µM, 10.6 µM, 14.1 µM, 17.7 µM, 26.5 µM, or 35.3 µM (based on the UPy-LL concentration, detailed explanation in Appendix A). After 24 h of incubation, the medium was removed, and the cells were washed with PBS buffer. After adding 200 µL of culture medium enriched with 44 µM resazurin (three empty wells were filled as control samples without cells), the cells were incubated for 3.5 h at 37 °C. Amounts of 2 × 80 µL of resazurin-enriched medium were transferred to a flat black 96-well plate, and the fluorescence was measured with a Synergy^TM^ HT plate reader and Gen5^TM^ software (BioTek Instruments, Inc.) at λ_ex_ = 550 nm and λ_em_ = 584 nm. Results were presented normalized to the fluorescence of the negative control (cells that were not in contact with the nanoparticles or the AMP).

### 3.4. Live/Dead Staining

The cell viability of both cell types in combination with BIONs, ION@PGA, ION@UPy-NH_2_, ION@UPy-NH_2_@UPy-LL, and free UPy-LL was determined by live/dead staining. An amount of 50 µL of cell suspension was seeded (with a seeding density of 2.5 × 10^5^ cells/cm^2^ for the THP-1 cells and 2.5 × 10^4^ cells/cm^2^ for the HK-2 cells) into a 15-well Ibidi slide (*n* = 3). The particles were sterilized as described above in Section 3.3. After the cell’s attachment (HK-2 overnight, THP-1 with PMA-induced medium for 48 h), the respective medium was exchanged for a medium containing 0.05 g L^−1^ or 0.50 g L^−1^ BIONs, ION@PGA, and ION@UPy-NH_2_ or 5.40 µM, 10.6 µM, or 14.1 µM ION@UPy-NH_2_@UPy-LL or UPy-LL. The cells were incubated for 24 h and washed with PBS (2×), incubated with 10 µM calcein AM and 10 µM propidium iodide-enriched medium (30 min, 37 °C), and washed with PBS (2×). Imaging acquisition of the cells was performed using a Zeiss Axio Observer 7 with a 10× objective.

### 3.5. Internalization

The internalization experiments of ION@UPy-NH_2_ containing UPy-Cy5 (Appendix A) were prepared by adding 2 µL of a 1.25 g L^−1^ UPy-Cy5 solution in chloroform to 100 µL of an 11.6 g L^−1^ ION@UPy-NH_2_ suspension in water. The particles were shaken at 30 °C and 1500 rpm for 6 h and washed by magnetic decantation (3×) to ensure the binding. They were sterilized as described before and diluted to a 0.1 g L^−1^ suspension in the respective medium. An amount of 50 µL of THP-1 or HK-2 cells was seeded onto a 15-well Ibidi slide (*n* = 3). After incubation with ION@UPy-NH_2_/UPy-Cy5 for 5 min, 2 h, 24 h, and 48 h, the cells were washed with PBS (2×), fixed with 15 µL of 10% formaldehyde solution at room temperature (25 °C), and washed with PBS (2×). The cells were sequentially stained with Phalloidin Alexa 488 staining solution (1:300, 30 min, 25 °C) for the membrane and 4′,6-Diamidin-2-phenylindol solution (DAPI, 2:1000, 10 min, RT) for the nuclei in PBS, respectively. After washing (2× PBS), imaging was performed with a Zeiss Axio Observer 7 with a 40× objective.

### 3.6. UPy-LL Binding

To combine ION@UPy-NH_2_ with UPy-LL, 1.0 mg of UPy-LL was dissolved in 176.5 µL 99:1 MeOH:1× PBS pH 7.4 (gibco) containing 2 mM 2-hydroxyethylpiperazine-N′-2-ethane sulfonic acid (HEPES) [28]. An amount of 28.7 µL of this solution was combined with 1.12 mL of 0.56 g L^−1^ ION@NH_2_ solution in PBS. The particles were shaken at 1000 rpm overnight. The particles were sterilized under UV light for 30 min, and the buffer was changed by magnetic decantation.

To analyze the binding behavior, different ION@UPy-NH_2_ concentrations (8 g L^−1^, 6 g L^−1^, 4 g L^−1^, 2 g L^−1^) in PBS were mixed 1:1 with various UPy-LL solutions (2 g L^−1^, 1 g L^−1^, 0.75 g L^−1^, 0.5 g L^−1^, 0.3 g L^−1^, 0 g L^−1^). The experiment was conducted in triplicate. The suspensions were mixed for one hour at room temperature at 1000 rpm. After magnetic decantation, the supernatants were photometrically measured at 280 nm, and the concentrations of unbound UPy-LL were calculated with the help of a calibration line.

The formation of the UPy-NH_2_ (bound to the IONs) and UPy-LL assemblies was examined by the Nile red (NR) encapsulation test on a Varian Cary Eclipse fluorescence spectrometer (Agilent Technologies, Santa Clara, CA, USA). A 0.1 g L^−1^ ION@UPy-NH_2_ suspension was combined with 0 µL, 0.08 µL, 0.16 µL, 0.32 µL, 0.64 µL, 1.28 µL, and 2.56 µL of UPy-LL as described above. After overnight incubation to let the fibers assemble, 0.53 µL of a 1 mM Nile red solution in MeOH was added (NR to UPy-LL 10:1). The particles were shaken at 400 rpm at room temperature for 20 min and then measured five times using a quartz cuvette (emission 565–800 nm, excitation 550 nm).

### 3.7. Antimicrobial Activity

The antimicrobial activity of ION@UPy-NH_2_, ION@UPy-NH_2_@UPy-LL, and free UPy-LL was analyzed with (RH)_4_-GFP-expressing *E. coli* (BL21, DE3) performed in a manner analogous to the method stated in Turrina et al. [17]. ION@UPy-NH_2_ in 50 mM PBS at pH 7.4 was diluted to 0, 0.1 g L^−1^, 0.3 g L^−1^, 0.5 g L^−1^, 0.8 g L^−1^, 1.0 g L^−1^, 2.0 g L^−1^, 3.0 g L^−1^, 4.0 g L^−1^, and 5.0 g L^−1^. After dissolving, probes containing 0.0 µM, 2.2 µM, 4.4 µM, 8.9 µM, 18 µM, 35 µM, 54 µM, and 106 µM of the free UPy-LL were prepared. After binding, ION@UPy-NH_2_@UPy-LL was diluted to similar UPy-LL concentrations (reaching a maximal particle concentration of 1.17 g L^−1^). During the experiment, all samples were finally diluted 1:10.

## 4. Conclusions

ION@UPy-NH_2_ is an interesting drug carrier material combing the favorable properties of IONs and supramolecular self-assembling systems. The UPy network forms a shell around the iron oxide core and small agglomerates, and it forms fibers. ION@UPy-NH_2_ and its precursor showed high cytocompatibility in HK-2 cells and a concentration-dependent effect on the metabolic activity of macrophagic THP-1 cells. The particles were internalized faster by THP-1-derived macrophages and were fully internalized after 24 h by both cell types. ION@UPy-NH_2_ binds UPy-LL with high efficiency (99%), inducing the reassembly and formation of smaller fibers. Attached to the drug carrier, the antimicrobial activity of UPy-LL is improved, leading to a MIC of 1.77 µM in *E. coli*. The combined system improved the cytocompatibility from 5.4 µM to 10.6 µM of the antimicrobial peptide for HK-2 cells, while it reduced the metabolic activity of THP-1-derived macrophages. The experiments demonstrated that ION@UPy-NH_2_ can be easily combined with UPy–drug molecules and could be used as a magnetically controlled drug delivery system for the antimicrobial peptide LL. The usage of low concentrations and improved cytocompatibility makes the AMP applicable.

## Figures and Tables

**Figure 1 ijms-24-14649-f001:**
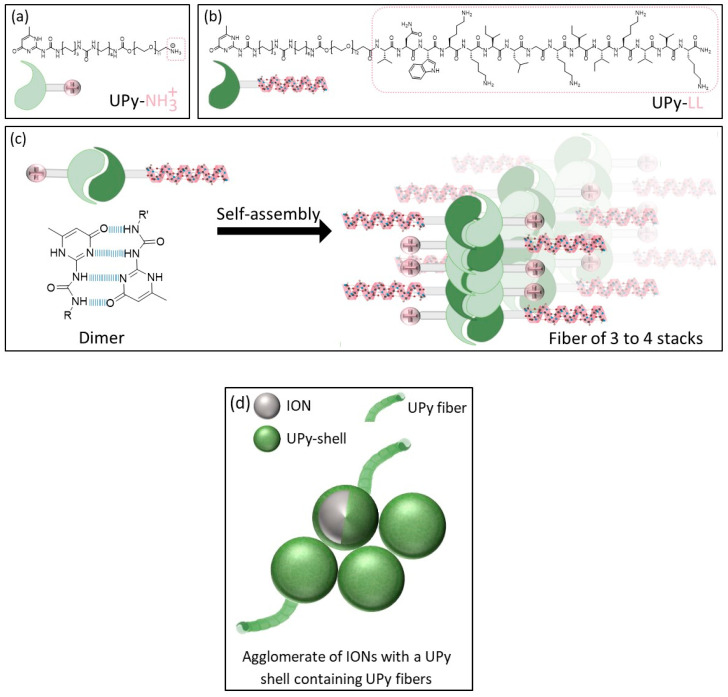
Schematic illustration of (**a**,**b**) the UPy unities (UPy-NH_2_, UPy-LL) forming dimers, stacks, and fibers, (**c**) fiber formation, and (**d**) schematic illustration of agglomerated IONs with a UPy shell containing UPy fibers [27].

**Figure 2 ijms-24-14649-f002:**
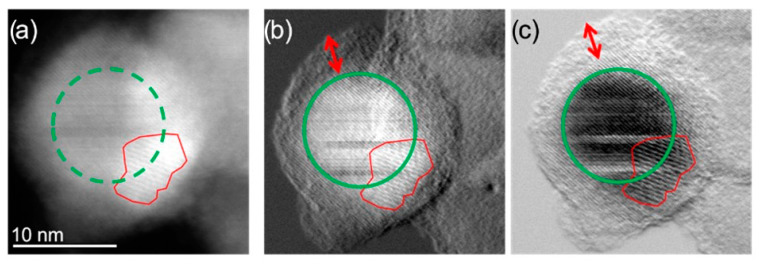
(**a**) HAADF-STEM micrographs of the same single ION@UPy-NH_2_ indicating the crystalline NP core (green circle) and an overlapping smaller particle creating Moiré fringes (red outline); (**b**) DPC_x_ (A–C) and (**c**) DPC_y_ (B–D) images showing more clearly the NP core and the amorphous coating. (A–C) and (B–D) indicate different detector segments. The red arrow indicates the coating thickness.

**Figure 3 ijms-24-14649-f003:**
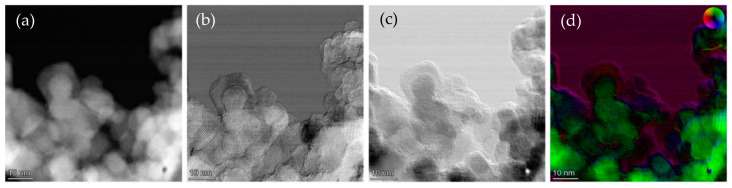
(**a**) HAADF-STEM micrographs of ION@UPy-NH_2_ agglomerates; (**b**) DPC_x_ (A–C) and (**c**) DPC_y_ (B–D) images; and (**d**) iDPC color display.

**Figure 4 ijms-24-14649-f004:**
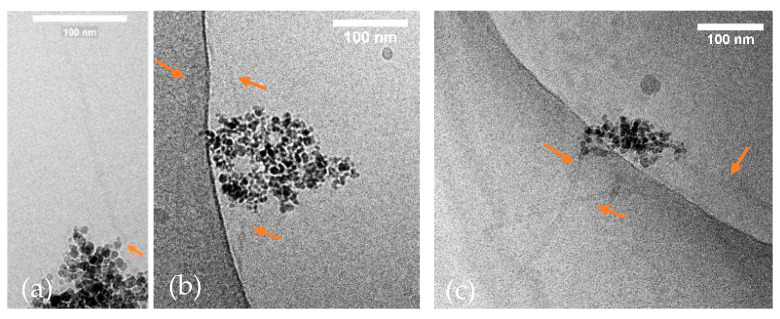
(**a**,**b**) Cryo-TEM micrographs of ION@UPy-NH_2_, and (**c**) ION@UPy-NH_2_@UPy-LL (35.3 µM UPy-LL were bound to 0.1 g L^−1^ ION@UPy-NH_2_). The fibers are marked with orange arrows.

**Figure 5 ijms-24-14649-f005:**
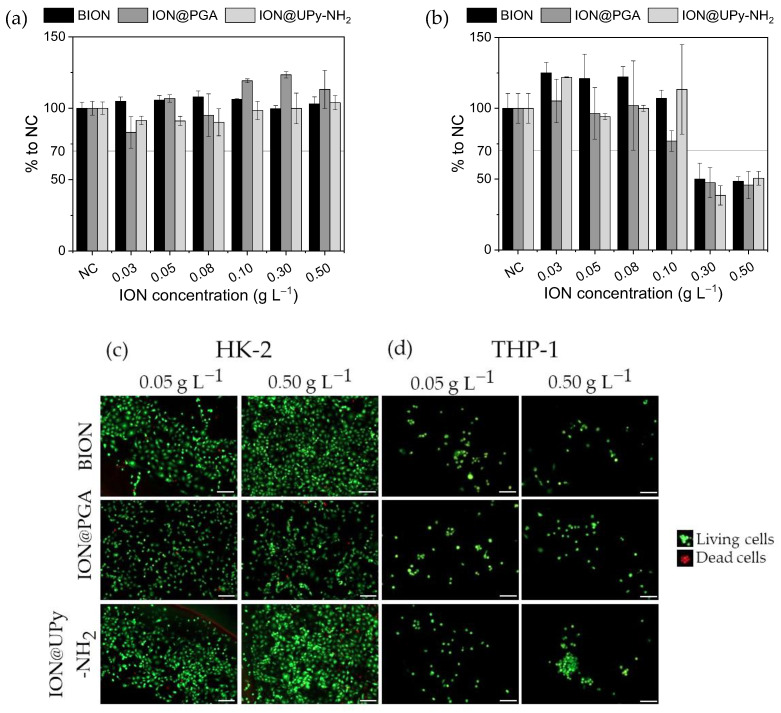
Cytocompatibility examination of ION, ION@PGA, and ION@UPy-NH_2_ by resazurin assay for (**a**) HK-2 cells and (**b**) THP-1 cells, and live/dead staining of (**c**) HK-2 and (**d**) THP-1 cells. The living cells were stained green, and the dead cells were stained red. Scale bars represent 200 µm.

**Figure 6 ijms-24-14649-f006:**
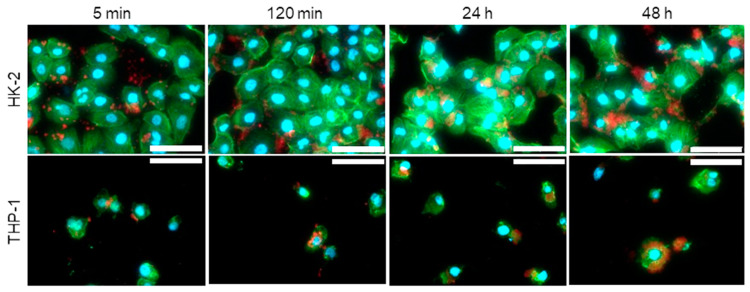
Fluorescent micrographs of internalization of ION@UPy-NH_2_ by HK-2 and THP-1 cells after 5 min, 120 min, 24 h, and 48 h. The ION@UPy-NH_2_ were labeled with 0.025 g L^−1^ UPy-Cy5 (red, Appendix A), and the nuclei (DAPI) and cell skeleton (Phalloidin Alexa 488) of the cells were stained blue and green, respectively. Scale bars represent 30 µm. All samples were examined under the same settings.

**Figure 7 ijms-24-14649-f007:**
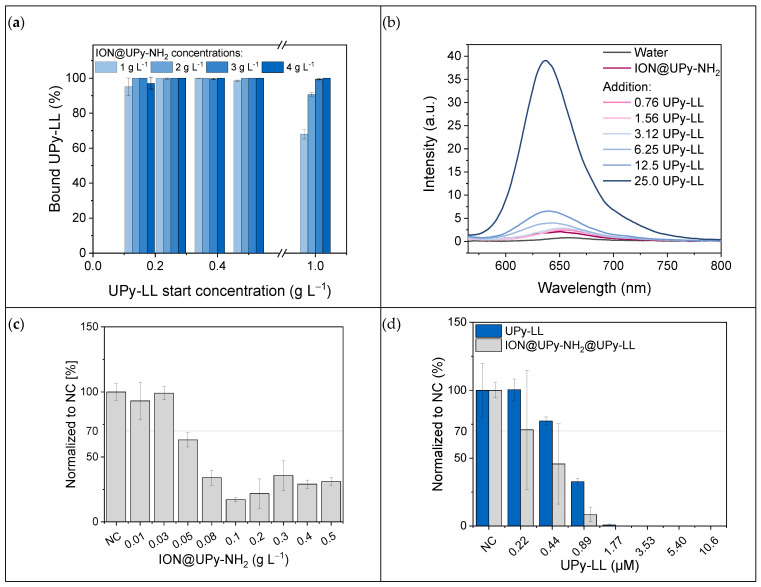
(**a**) Binding of UPy-LL to different concentrations of ION@UPy-NH_2_ and (**b**) Nile red assay of ION@UPy-NH_2_@UPy-LL with different UPy-LL concentrations. Growth studies with *E. coli* and (**c**) different concentrations of ION@UPy-NH_2_ and (**d**) various amounts of UPy-LL and ION@UPy-NH_2_@UPy-LL complexes.

**Figure 8 ijms-24-14649-f008:**
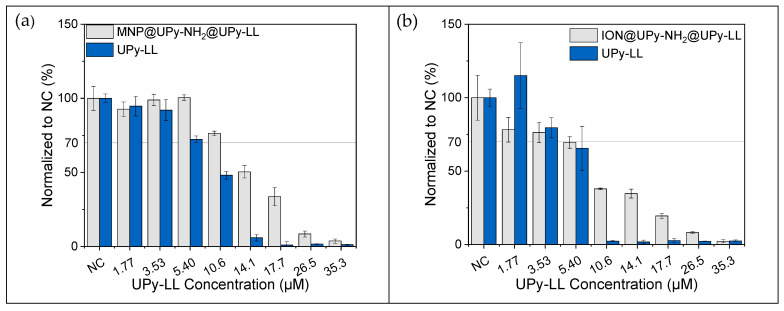
Cytocompatibility examination by resazurin assay for UPy-LL and ION@UPy-NH_2_-UPy-LL in (**a**) HK-2 cells and (**b**) THP-1 cells, and live/dead staining of (**c**) HK-2 and (**d**) THP-1 cells. The living cells are colored green and the dead cells red. Scale bars represent 200 µm.

## Data Availability

The datasets generated and/or analyzed during the current study are available from the corresponding author on reasonable request.

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
