# Peer review of "Iron Oxide Nanoparticles with Supramolecular Ureido-Pyrimidinone Coating for Antimicrobial Peptide Delivery"

_ijms, 2023, doi:10.3390/ijms241914649_

Round 1
Reviewer 1 Report
The work by Turrina et al. describes the Iron oxide nanoparticles with supramolecular ureidopyrimidinone coating for antimicrobial peptide delivery, as the title states.
The presented results are of limited novelty as they very little augment to the already published data by the same authors, namely:
Bare Iron Oxide Nanoparticles as Drug Delivery Carrier for the Short Cationic Peptide Lasioglossin. Pharmaceuticals 2021, 14, 405. https://doi.org/10.3390/ph14050405
Turrina, C., Dankers, P. Y. W., Berensmeier, S., & Schwaminger, S. (2022). Iron oxide nanoparticles with supramolecular ureidopyrimidinone coating. Current Directions in Biomedical Engineering, 8(2), 13-16. https://doi.org/10.1515/cdbme-2022-1004
Turrina, C.; Milani, D.; Klassen, A.; Rojas-González, D.M.; Cookman, J.; Opel, M.; Sartori, B.; Mela, P.; Berensmeier, S.; Schwaminger, S.P. Carboxymethyl-Dextran-Coated Superparamagnetic Iron Oxide Nanoparticles for Drug Delivery: Influence of the Coating Thickness on the Particle Properties. Int. J. Mol. Sci. 2022, 23, 14743. https://doi.org/10.3390/ijms232314743
The latter two are cited in the refeferences under numbers 16 and 42, which are however incomplete.
Although the authors use and comment on their previously published data, the presentation of the new results in this manuscript is rather messy. I attach the pdf file of it where I've highlighted words/phrases that have to be corrected. Here I list shortly some of the major corrections that are needed.
1. The subchapter numberring is wrong and must be corrected.
2. When using Results and discussion as way to present the data, please start each paragrpah with your current results and give discussion afterwards on previously published data but only those that are closely relevant to yours. So, avoid general discussion on broad topic that only makes the reader even more confused. Clearly compare your current result with your previously published on the similar systems and try to highlight the seen differences.
3. Is Figure 4 necessary? and if Yes, shouldn't it go in the comments of subsection 3.2 and thus appear after figure 5 ? If you want the fibers in the imagaes to be seen remove the orange elipces and use small arrows instead.
4. Please, avoid general statements when describing your or others results - they don't say anything, e.g. "showed good cell viability for higher concentrations (Figure 5d)." line 193 / The effect is smaller for live/dead staining. Here only a few dead cells are visible for HK-2 and slightly more for THP-1. The small amount of dead cells could come through a loss during washing, in all experiments the same amount of cells was seeded. The cytotoxic effect of UPy-LL gets moderated by binding to the particles, and ION@UPyNH2@UPy-LL showed better cytocompatibility to HK-2 cells than to the macrophagic ones (lines 308-312) / and in several other palces such general and confusing statements are not helpful.
5. In Fig. 5 do you see red cells ? - I cannot. Back to the general statement at line 192-193 - which of your data do you trust more? - the staining experiment or the resazurine assay.
6. For the resazurine assay - do you need control (empty, with no cell), and then NC negative control that is untreated cells? Usually viability data are presented as survival % of the untreated cells and it is not called negative control, and there is no need to show them in the graph as a bar with 100 % viability
7. The first two sentences in the Conclusions are not true - these are observations already described in ref. 42. Please, focus on the new findings in your current work.
In my opinion this manuscript should be re-written to make it readable and comprehendable given there are many prior data that have been published on similar systems but in different combinations. What is the novelty in the current results should be clearly stated and clearly discussed in comparison with the ones already published.
English language must be carefully revised - my highlights are just for the obviuos mistakes and missunderstandings.

English language must be carefully revised - my highlights in the attached pdf file are just for the obviuos mistakes and missunderstandings.
Author Response
the answer is attached

Reviewer 2 Report
There are following issues on the article:
1. There are several typos in the text
2. Significant questions on the physico-chemical analysis
3. There is no statistical data processing
4. There are low resolution micrographs without scalebars
5. There is no quantitative calculation of both the proportion of dead cells and the colocalization of nanoparticles
6. Significant questions on the cytocompatibility and the antimicrobial activity analysis
7. Significant questions on the methods
8. Significant questions on the conclusions
All issues are described in detail in the attached pdf-review

Author Response
the answer is attached

Reviewer 3 Report
Turrina et al. merge supramolecular engineering with nanotechnology to enhance antimicrobial activity and diminish toxicity. Given that the synthesis groundwork is covered in a prior article, a brief overview of the synthesis route and coating processes would assist readers in understanding the ION@UPy-NH2 platform's application. Specific comments for improvement include:
1. Line 70: Elaborate on the concept of supramolecular engineering in the Introduction's second paragraph. Define it, cite prominent examples, and enumerate its advantages.
2. Figure 1: The figure should be segmented into distinct panels (a, b, c, d) with each panel discussed in detail within the legend. The use of "left/right" is ambiguous. Additionally, clarify every molecular notation, encompassing the LL peptide, the UPy molecule, and the amine group.
3. Line 108: After mentioning a two-step functionalization, details on the initial APTS functionalization on IONs should follow. Subsequently, introduce the abbreviation ION@UPy-NH2.
4. Despite prior coverage on synthesis and characterization, it would be beneficial to outline the synthesis route briefly in both the results and methods sections, given its pivotal role.
5. Lines 138-139: Elucidate the abbreviations HAADF-STEM and DPC imaging. Why were different imaging techniques employed? Establish their significance in inspecting ION morphology.
6. Figure 2b, c: Insert scale bars.
7. Maintain consistency in the manuscript. For instance, on lines 155-156, 163-164, pinpoint the locations of (A-C) and (B-D) images. Clearly delineate the red arrows' significance in the legend. On line 187, reference figure 5a, not 4a. On line 232, adjust the scale bars unit to µm. Rectify references on lines 275, 281, and 295 to figures 7c, 7d, and 8a and b, respectively.
8. Figure 5d: The cell density for THP-1 appears sparse. Given the 50% cell death rate at 0.5 g/L as per Figure 5b, the scant red signal in Figure 5d is perplexing.
9. Figure 6: Integrate an additional panel, possibly a bar graph, to assess the normalized red signal within cells. While most signals seem to originate from the cell's skeleton, it doesn't appear linked to ION endocytosis.
Author Response
the answer is attached

Round 2
Reviewer 1 Report
Authors answered all my comments and question and can recommend the manuscript for publication. If they can change the graphical abstract to something more meaningful it would be better.
The language is fine in general.
Author Response
Thank you
Reviewer 2 Report
I propose to publish the revised manuscript in this form
Author Response
Thank you
Reviewer 3 Report
1. In the legends accompanying Figures 2 and 3, the notations “b) DPCx (A-C), and c) DPCy (B-D)” are mentioned. Could you please clarify what the representations "(A-C)" and "(B-D)" signify in this context?
2. It appears that Figure 5a and 5b are absent in the current version of the manuscript. Furthermore, the sparse red signaling observed in Figure 5d seems inconsistent with the previously indicated 50% cell mortality rate at a concentration of 0.5 g/L (as per the earlier version of Figure 5b). The discrepancy between the resazurin assay outcomes and the cell death staining needs to be addressed and justified, particularly explaining why these two methods would yield divergent observations.
3. For Figure 6, integrating an extra panel, perhaps a bar graph, to evaluate the normalized red signals in the cells would be beneficial. At the moment, the signals predominantly appear to emanate from the cell skeleton, which doesn't evidently correlate with ION internalization. The manuscript claims a substantial degree of internalization after 2 hours, increasing further at the 24-hour mark, based on observations with "ON@UPy-NH2". Could you please substantiate these claims with statistical evidence to unequivocally demonstrate an increase in signal over the timeframe mentioned?
4. The manuscript seems to hypothesize an endocytosis mechanism for uptake, a theme that is explored within the context. However, the current version lacks empirical data to firmly substantiate this theory. To robustly delineate the uptake pathway, I would suggest incorporating experiments utilizing a range of endocytosis inhibitors, complemented possibly with experiments at 4°C, to provide a comprehensive understanding of the mechanism at play.
Author Response
Thank you for your review. We attached a report answering your questions point-by-point.

Round 3
Reviewer 3 Report
The publication is in a good format and ready for publication.